# Treatment Regimens for Immunocompetent Elderly Patients with Primary Central Nervous System Lymphoma: A Scoping Review

**DOI:** 10.3390/cancers13174268

**Published:** 2021-08-24

**Authors:** Elisabeth Schorb, Lisa Kristina Isbell, Gerald Illerhaus, Gabriele Ihorst, Joerg J. Meerpohl, Kathrin Grummich, Blin Nagavci, Christine Schmucker

**Affiliations:** 1Department of Hematology, Oncology and Stem Cell Transplantation, Faculty of Medicine, University of Freiburg, 79106 Freiburg, Germany; lisa.isbell@uniklinik-freiburg.de (L.K.I.); gabriele.ihorst@uniklinik-freiburg.de (G.I.); 2Department of Hematology, Oncology and Palliative Care, Klinikum Stuttgart, 70174 Stuttgart, Germany; g.illerhaus@klinikum-stuttgart.de; 3Clinical Trials Unit, Faculty of Medicine, University of Freiburg, 79110 Freiburg, Germany; 4Institute for Evidence in Medicine, Medical Center & Faculty of Medicine, University of Freiburg, 79110 Freiburg, Germany; meerpohl@ifem.uni-freiburg.de (J.J.M.); grummich@cochrane.de (K.G.); nagavci@ifem.uni-freiburg.de (B.N.); schmucker@ifem.uni-freiburg.de (C.S.); 5Cochrane Germany, Cochrane Germany Foundation, 79110 Freiburg, Germany

**Keywords:** primary central nervous system lymphoma, elderly patients, conventional chemotherapy, high-dose chemotherapy, autologous stem cell transplantation

## Abstract

**Simple Summary:**

Most patients diagnosed with primary central nervous system lymphoma (PCNSL) are 60 years or older and tend to have a poor prognosis. Evidence to guide and optimize treatment choices for these vulnerable patients is limited. We performed a scoping review to identify and describe all relevant clinical studies investigating chemotherapies and combinations of chemotherapies (including high-dose chemotherapy followed by autologous stem cell transplantation (HCT-ASCT)) in elderly PCNSL patients. In total, we identified six randomized controlled trials, 26 prospective and 24 retrospective studies (with/without control group). While most studies investigated protocols based on ‘conventional’ chemotherapy treatment, data evaluating HCT-ASCT in the elderly were scarce, and the generalizability of the only RCT published is questionable. Considering the poor prognosis of these patients and their need for more effective treatment options, a thoroughly planned randomized controlled trial comparing HCT-ASCT with ‘conventional’ chemoimmunotherapy is urgently needed to evaluate the efficacy of HCT-ASCT.

**Abstract:**

Background: Most patients diagnosed with primary central nervous system lymphoma (PCNSL) are older than 60 years. Despite promising treatment options for younger patients, prognosis for the elderly remains poor and efficacy of available treatment options is limited. Materials and Methods: We conducted a scoping review to identify and summarize the current study pool available evaluating different types and combinations of (immuno) chemotherapy with a special focus on HCT-ASCT in elderly PCNSL. Relevant studies were identified through systematic searches in the bibliographic databases Medline, Web of Science, Cochrane Library and ScienceDirect (last search conducted in September 2020). For ongoing studies, we searched ClinicalTrials.gov, the German study register and the WHO registry. Results: In total, we identified six randomized controlled trials (RCT) with 1.346 patients, 26 prospective (with 1.366 patients) and 24 retrospective studies (with 2.629 patients). Of these, only six studies (one completed and one ongoing RCT (with 447 patients), one completed and one ongoing prospective single arm study (with 65 patients), and two retrospective single arm studies (with 122 patients)) evaluated HCT-ASCT. Patient relevant outcomes such as progression-free and overall survival and (neuro-)toxicity were adequately considered across almost all studies. The current study pool is, however, not conclusive in terms of the most effective treatment options for elderly. Main limitations were (very) small sample sizes and heterogeneous patient populations in terms of age ranges (particularly in RCTs) limiting the applicability of the results to the target population (elderly). Conclusions: Although it has been shown that HCT-ASCT is probably a feasible and effective treatment option, this approach has never been investigated within a RCT including a wide range of elderly patients. A RCT comparing conventional (immuno) chemotherapy with HCT-ASCT is crucial to evaluate benefit and harms in an un-biased manner to eventually provide older PCNSL patients with the most effective treatment.

## 1. Introduction

Primary diffuse large B-cell lymphoma of the central nervous system (PCNSL) is an orphan disease with an age-adjusted annual incidence rate of seven cases per million in the United States [1]. The Central Brain Tumor Registry of the United States (CBTRUS) 2011–2015 report estimates that PCNSL represents approximately 1.9% of all primary central nervous system (CNS) tumors and 6.3% of malignant CNS tumors [2]. Elderly patients (>60 years) are more commonly affected and the incidence is increasing [3].

More than 90% of PCNSL cases are of the diffuse large B-cell type [4] and the majority of which have ≥ 1 mutations in the NF-kB and B-cell receptor signaling pathways, such as CD79B, MYD88, TBL1XR1, CARD11, or CDKN2A [5]. Furthermore, PCNSL frequently show immunoglobulin rearrangement, specifically IGHV4-34 [6]. Based on their immunophenotype and gene expression, PCNSL in immunocompetent patients share features of late germinal center and activated post–germinal center B-cells [7,8,9,10].

PCNSL risk is highly elevated among patients with acquired immunosuppression [11]. In immunosuppressed patients, PCNSL is frequently associated with ineffective immunoregulation of EBV-associated B-cell proliferation and high rates of Epstein-Barr-Virus (EBV) positivity. Conversely, only 5–15% of immunocompetent PCNSL are EBV positive [12]. EBV-associated PCNSL significantly differs from EBV-negative disease as it is typically absent of CD79B and MYD88 mutations and is rarely ABC cell of origin [13]. Importantly, although the CNS is an immune-privileged niche under physiological conditions, PCNSL frequently contain a strong inflammatory response [14]. The extent to which age-specific differences might arise with regard to the characteristics described above has not been shown to date.

Unlike many brain tumors, the typical disease history of patients with PCNSL extends only over a short period of a few weeks. Frequently, patients are noted for personality changes, memory or language deficits, neuropsychiatric symptoms, or focal neurologic deficits. Less common initial symptoms include uveitis, seizures, or increased intracranial pressure [15]. The majority of PCNSL patients initially present with only a singular focus of lymphoma, whereas disseminated, multifocal forms of disease are much rarer [16] and approximately 15% of patients present with ocular involvement at initial diagnosis [17]. Older age should not obviate establishment of a diagnosis. Stereotactic biopsy is the standard of care to obtain a histological diagnosis [18].

PCNSL patients frequently suffer a high burden of disease with various neurological symptoms leading to rapid clinical deterioration and death if not immediately treated. Thus, patients with PCNSL in general, and in particular elderly patients, require high resources for obtaining optimal age- and comorbidity-adapted treatment management during their often multiple hospital stays [19]. Additionally, older patients have an inferior prognosis compared to younger patients [20,21] and are more seriously affected by treatment toxicity, especially neurotoxicity (following treatment with whole brain radiotherapy (WBRT)) accompanied by dementia, ataxia, gait disturbances, and incontinence) [22]. Thus, treatment decisions in elderly PCNSL patients must be individualized, taking into account pre-morbid performance status and comorbidities. Importantly, age alone should not be a barrier for delivery of an intensive treatment regimen if patients appear to have adequate physiological fitness [23].

The current treatment standard for newly diagnosed (elderly) PCNSL patients is high-dose methotrexate (HD-MTX)-based immuno-chemotherapy [24,25,26,27,28]. Consolidation treatment is typically used to prolong remission after induction therapy. Commonly used consolidation strategies comprise non-myeloablative chemotherapy [29], WBRT, or high-dose chemotherapy followed by autologous stem cell transplantation (HCT-ASCT) [30,31]. The rationale behind HCT-ASCT in PCNSL is to increase the treatment concentration by a multiple factor to support diffusion across the blood brain barrier, resulting in penetration into the CNS—which cannot be achieved with conventionally dosed therapy. In younger patients, HCT-ASCT has become the most widely accepted consolidation approach. Two RCTs [32,33] and various single arm studies [34,35,36,37,38,39] investigated thiotepa-based HCT-ASCT protocols in PCNSL patients younger than 65 and 70 years of age, with 4-year overall survival (OS) rates over 80% [32]. However, elderly (PCNSL) patients often fail to receive optimal treatment due to lacking evidence from clinical studies.

Recently, HCT-ASCT has been increasingly used in selected elderly PCNSL patients who are able to tolerate aggressive systemic chemotherapy. Nevertheless, an optimal approach regarding treatment intensification-particularly in elderly PCNSL patients- needs to be established. To overcome this challenge it is important to identify the eligible elderly patient population with newly diagnosed PCNSL tolerating more intense and shorter HCT-ASCT treatment protocols and hence benefit from this treatment option (i.e., patients who improve in efficacy outcomes without increasing toxicity).

We aimed to summarize the current study pool available evaluating different types and combinations of chemotherapy with a special focus on HCT-ASCT in elderly PCNSL patients by using the methods of a scoping review [40]. The results of this scoping review are an important part of the conceptual development phase for a planned prospective international, multicenter randomized controlled trial investigating the efficacy and safety of “HCT-ASCT in comparison to conventional chemotherapy with the rituximab-MTX-procarbazine (R-MP) protocol followed by procarbazine maintenance in elderly patients with newly diagnosed PCNSL” (prospective registration identifier of the clinical trial: DRKS DRKS00024085).

The scoping review will explore the different chemotherapy-containing treatment protocols in elderly PCNSL patients investigated in RCTs and non-randomized clinical studies (including one-arm studies), the exact eligibility criteria of the populations included in the studies, the diagnostic methods used for defining eligibility for different treatment approaches, and the outcomes reported. Outcomes of interest will be progression-free survival (PFS), event-free survival (EFS) and overall survival (OS), remission rates and toxicity parameters. The results of the scoping review will highlight areas where clinical studies are lacking. Furthermore, the review will support us to finalize and adjust the research protocol (including the design and methodology) of the planned randomized controlled trial by our team (PRIMA-CNS trial).

## 2. Materials and Methods

This scoping review was registered at OSF (registration DOI 10.17605/OSF.IO/ZPCNU). Comprehensive systematic literature searches for relevant studies were conducted according to PRESS (Peer Review of Electronic Search Strategies) guidelines [41]. These searches were performed by an information specialist without any date restrictions. The initial systematic search with an explicit focus on HCT-ASCT in PCNSL was conducted on 29 May 2020 in the electronic data sources Medline, Medline Daily Update, Medline In Process & Other Non-Indexed Citations, Medline Epub Ahead of Print (via Ovid)), Web of Science Core Collection (Science Citation Index-EXPANDED), Cochrane Library (via Wiley), and ScienceDirect (via Elsevier). The second search (with a broader focus regarding the intervention, i.e., with the focus on any type and combination of chemotherapy) was conducted on 8 September 2020 in Medline, Medline Daily Update, Medline In Process & Other Non-Indexed Citations, Medline Epub Ahead of Print (via Ovid; Appendix A). The reason for the second broader search was that potentially relevant studies were not identified by limiting the intervention to HCT-ASCT only. Therefore, we conducted this additional search focused on any type or combination of chemotherapy in elderly PCNSL patients (our population of interest). Searches for ongoing or unpublished completed studies were performed in ClinicalTrials.gov (www.clinicaltrials.gov, accessed on 5 June 2020) and the German study register (www.drks.de, accessed on 5 June 2020). We used relevant studies and/or systematic reviews to search for additional references via the Pubmed similar articles function (https://www.nlm.nih.gov/bsd/disted/pubmedtutorial/020_190.html, accessed on 5 June 2020) and forward citation tracking using the Web of Science Core Collection. Furthermore, reference lists of relevant studies and systematic reviews were scanned for potentially relevant studies not captured by other searches.

Titles and abstracts of the references identified by the searches were screened by one reviewer (B.N.) and full texts of all potentially relevant articles were obtained. Full texts were checked for final eligibility and reasons for exclusions were documented. The screening process was conducted in Covidence (www.covidence.org, accessed on 5 June 2020).

Studies including immunocompetent PCNSL patients aged 60 years or older (≥60) receiving any therapy line were included. Studies including younger patients (aged < 60 years) or patients with mixed ages (</≥60 years) without providing subgroup analysis for older ages (≥60) as well as those including immunocompromised patients with PCNSL were excluded. All types, doses and combinations of chemotherapy-based treatment regimen including HCT-ASCT were considered.

Randomized controlled trials, non-randomized studies of interventions including studies in which individuals are allocated to different interventions using methods that are not random; observational studies and single arm studies were considered, whereas case reports, review articles, work without peer-review and results reported in abstract form only were excluded. No exclusion criteria regarding study duration were applied.

Key study data including, characteristics of the participants, characteristics of the intervention, characteristics of the comparator, outcomes and their definitions were extracted and relevant information tabulated. Data from each included study were extracted by 1 reviewer (B.N.) and checked by a second (C.S.). Disagreements were resolved through discussion until consensus was reached.

## 3. Results

Overall, 1335 records were identified by our systematic searches, of which 234 were considered for full-text assessment. In total, 56 studies corresponding to 61 publications full-filled the inclusion criteria for the scoping review. The PRISMA flow diagram (Figure 1) outlines the screening and selection process of these articles. Table 1, Table 2 and Table 3 and Appendix A present the key characteristics and main outcomes of the identified six RCTs, two prospective non-randomized studies (with control group), 24 prospective single arm studies (including one protocol for an ongoing study) and 24 retrospective studies (seven with control group and 17 single arm studies).

### 3.1. Randomized Controlled Trials (RCTs)

#### 3.1.1. Key Characteristics of Randomized Controlled Trials (RCTs)

The key characteristics of the six RCTs (five completed, one ongoing [MATRix trial]) are displayed in Table 1.

##### RCTs Comparing Different Types of Chemotherapy (N = 2)

Setting, follow-up: Two RCTs compared different types of chemotherapy; the multicenter and multinational, open-label phase III study conducted by Bromberg et al. and the multicenter Phase II study authored by Omuro et al. The latter recruited 95 patients between the years 2007 and 2010 and was conducted at 13 centers in France with a median follow-up time of 32 months (interquartile range [IQR] 26–36) [43]. Bromberg et al. randomized 200 participants across 23 centers in the Netherlands, Australia, and New Zealand. The study recruited participants between 2010 and 2016 with a median follow-up time of 32.9 months (IQR 24–52) [44].

Definition of patient population: Both RCTs included immunocompetent patients with newly diagnosed PCNSL confirmed histologically and/or with neuroimaging. While the study of Omuro et al. is the only trial specifically designed for elderly PCNSL patients (60 years or older, Karnofsky Performance Status (KPS) of 40 or more), Bromberg et al. included (younger) patients up to the age of 70 years (Eastern Cooperative Oncology Group Performance Status (ECOG PS) between 0 and 3)). In Omuro et al., median age was 73 years (range 60–85) in the intervention and 72 years (range 60–84) in the control group. Bromberg et al. reported a median of 61 years (range 55–67) in the intervention and 61 years (range 56–66) in the control group, respectively.

Treatment protocol: In Bromberg et al., all patients were treated with a chemotherapy backbone comprising two 28-day-cycles of HD-MTX 3 g/m^2^/day for 2 days, carmustine 100 mg/m^2^ once, teniposide 100 mg/m^2^/day for 2 days, prednisone 60 mg/m^2^/day for 6 days and consolidating cytarabine (AraC) 2 × 2 g/m^2^ for 2 days. Half of the patients (N = 100) were randomized to additionally receive the anti-CD20 antibody rituximab 375 mg/m^2^/for 4 days in cycle 1 and 2 days in cycle 2 (intervention group). Importantly, the treatment regimen of younger patients differed from that of elderly patients with regard to consolidation treatment. In responding patients, consolidating WBRT with 20 fractions of 1.5 Gray (Gy) with an additional integrated boost to the tumor bed of 20 fractions of 0.5 Gy in patients with only partial remission (PR) was applied in a subgroup of patients aged ≤ 60 years whereas patients > 60 years did not receive additional treatment [44].

Within their randomized trial specifically designed for elderly PCNSL patients, Omuro et al. compared two different chemotherapy combination regimen of different intensity. Forty-eight patients received three 28-day-cycles of HD-MTX 3.5 g/m^2^/day for two days in combination with the oral alkylating agent temozolomide 150 mg/m^2^/day for five days (intervention group). Forty-seven patients received three 28-day-cycles of polyagent chemotherapy with HD-MTX 3.5 g/m^2^/day for 2 days, procarbazine 100 mg/m^2^/day for 8 days, vincristine 1.4 mg/m^2^/d for 2 days and consolidating AraC 3g/m^2^/day for 2 days (control group) [45].

##### RCT Comparing Chemotherapy with WBRT vs. Chemotherapy Alone (N = 1)

Setting, follow-up: The multicenter phase III trial of the German PCNSL Study Group (G-PCNSL-SG-1 trial) was conducted across 75 German centers. Participants were recruited between the years 2000 and 2009 [46]. In the context of this trial, there were a total of five reports published [24,46,47,48,49], resulting in different follow-up times with a maximum median follow-up time of 81.2 months. A subgroup analysis for patients > 60 years was provided. Additionally, within a post-hoc analysis, the study population of the G-PCNSL-SG-1 trial was divided arbitrarily into two age groups with a cut-off of 70 years or older. Within this post-hoc analysis, outcome data for 126 patients > 70 years was reported [24].

Definition of patient population: Overall, 551 immunocompetent patients with newly diagnosed, histologically proven PCNSL were included without age limitations. However, patients with KPS less than 50% for reasons not related to PCNSL, and of less than 30% for reasons related to PCNSL were excluded. Median age of the two groups was 62 (SD 10.8) (intervention group) and 61 years (SD 11.6) (control group).

Treatment protocol: Patients included in the G-PCNSL-SG-1 trial received six 14-day cycles of HD-MTX 4 g/m^2^ (in the further course of the trial ifosfamide 1.5 g/m^2^/d for 3 days was added) with or without randomly assigned WBRT consolidation treatment with a total dose of 45 Gy. Those patients who were allocated to treatment without WBRT and who did not achieve complete remission (CR) were given four cycles of high-dose AraC (HD-AraC) 3 g/m^2^ twice daily for 2 days.

##### RCTs Comparing Chemotherapy with WBRT vs. WBRT Alone (N = 1)

Setting, follow-up: The phase II study by Mead et al. was conducted across 12 centers in the United Kingdom between 1988 and 1995. Recruitment was stopped prematurely through poor accrual. Fifty-three previously untreated, immunocompetent adult patients with pathologically proven PCNSL were randomized between WBRT with or without previous polychemotherapy treatment. Each randomized group included patients > 60 years (WBRT + chemotherapy N = 17; WBRT N = 3). Median follow-up was 60 months (range 12–108) [50].

Definition of patient population: In the trial by Mead et al. adults without any limitations regarding age and ECOG PS were included but patients with neurologic status (Medical Research Council Neurological Scale of 3 or less) were excluded.

Treatment protocol: Patients were randomly assigned to receive six 21-day cycles of post-surgery and -radiotherapy chemotherapy treatment with cyclophosphamide 750 mg/m^2^, doxorubicin 50 mg/m^2^, vincristine 1.4 mg/m^2^ and prednisone 20 mg/d for 5 days (CHOP) vs. no additional chemotherapy treatment.

##### RCTs Evaluating HCT-ASCT (N = 2, Results Not Applicable for Elderly or Pending)

Setting, follow-up:

The phase II International Extranodal Lymphoma Study Group (IELSG32) trial was the first reported trial directly comparing HCT-ASCT with WBRT. The trial (by Ferreri et al.) was conducted across 53 centers in five European countries [32,51]. From 2010 to 2014, a total of 227 patients with newly diagnosed PCNSL up to 70 years were included. Median age across the groups ranged between 58 and 57 years, with a median follow-up time of 40 months. The first randomization compared different induction chemotherapy regimens [51]. The second randomization compared HCT-ASCT with WBRT and considered 118 patients (see below) [32].

The subsequent ongoing MATRix/IELSG43 trial is conducted in 5 European countries [29]. Within this randomized phase III RCT, 220 patients are randomized (after induction treatment with the MATRix protocol) between consolidating HCT-ASCT and ‘conventional’ consolidating treatment. Recruitment was completed in August 2019, follow-up is ongoing and results are expected in 2022.

Definition of patient population: In the IELSG32 and in the MATRix/IELSG43 trial, patients were eligible after the following criteria: (i) ≤65 years with ECOG PS ≤ 3 and (ii) up to the age of 70 years only with ECOG PS ≤ 2.

Treatment protocol: Within the IELSG32 trial by Ferreri et al. induction treatment consisted of different intensity (depending on the treatment arm). Those patients randomly assigned to arm A received four 21-day-cycles of HD-MTX 3.5 g/m^2^ once and AraC 2 g/m^2^ twice daily for 2 days, patients assigned to arm B additionally received the anti-CD20 antibody rituximab 375 mg/m^2^ twice, whereas patients assigned to arm C received the combination of arm B plus thiotepa 30 mg/m^2^ once (so called MATRix regimen). Patients achieving stable or responsive disease were again randomly assigned to consolidation treatment with either WBRT (36 Gy with an additional nine Gy tumor-bed boost in patients with partial response, N = 59 patients) or HCT with carmustine 400 mg/m^2^ once and thiotepa 5 mg/kg twice a day for 2 days followed by ASCT (N = 59 patients) (second randomization). Within the subsequent (still ongoing) MATRix/IELSG43 trial, four 21-day-cycles of the MATRix regimen were administered followed by randomization between consolidating HCT-ASCT with carmustine 400 mg/m^2^ once and thiotepa 5 mg/kg twice a day for 2 days or conventional consolidating treatment with rituximab 375 g/m^2^ once, dexamethasone 40 mg/d for 3 days, etoposide 100 mg/m^2^/d for 3 days, ifosfamide 1500 mg/m^2^/d for 3 days and carboplatin 300 mg/m^2^ once (R-DeVIC protocol) in responding patients.

#### 3.1.2. Outcomes of the Randomized Controlled Trials (RCTs)

Table 2 provides an overview of the outcomes considered and main results of the five completed randomized trials.

##### RCTs Comparing Different Types of Chemotherapy (N = 2)

Bromberg et al. reported event-free survival (EFS) as the primary outcome. Events were defined as the absence of (unconfirmed) complete remission (CR) at the end of protocol treatment, or relapse or death after previous (unconfirmed) CR (according to the International PCNSL Collaborative Group (IPCG) Response Criteria [18]). Overall, the authors reported that outcomes were not different within the two treatment groups. For example, one-year EFS was 52% (95% CI 42–61) in the patients who additionally received rituximab (intervention group) and 49% (95% CI 39–58) in the chemotherapy arm without rituximab (control group). Notably, additional WBRT after chemoimmunotherapy was only administered to younger patients (≤60 years). Subgroup analysis showed that younger patients (≤60 years) had better results in terms of EFS when rituximab was added (intervention group; median EFS 59.9 months (95% CI 41.4–not reached)) compared to the control group (median EFS 19.7 months (95% CI 6.5–not reached) (HR 0.56, 95% CI 0.31–1.01, *p* = 0.054). In the subgroup for older patients (>60 years) no difference between the treatment arms was observed [44]. However, these findings have to be interpreted with caution, as consolidation strategy with WBRT varied significantly between these age groups.

The RCT by Omuro et al. reported one-year progression-free survival (PFS) as primary outcome. PFS was defined as time to progression (determined by local investigators) or death. Secondary outcomes were overall survival (OS), toxicity, objective response, quality of life (QoL) and neuropsychological evaluation. One-year PFS was 36% (95% CI 22–50) in both groups, but OS and response rates favored the more intensive treatment group (control group). No differences in toxicity between the study groups were observed. Importantly, QoL improved across most domains in comparison to baseline in both groups without evidence of late neurotoxicity (using prospective neurocognitive assessments).

##### RCT Comparing Chemotherapy with WBRT vs. Chemotherapy Alone (N = 1)

The G-PCNSL-SG-1 trial reported OS defined as time between randomization and death or the date when “last seen alive”. The publication of Thiel et al. reported no significant differences regarding the primary endpoint OS, but non-inferiority was not proven (non-inferiority margin of 0.9) when WBRT was omitted from chemotherapy. A subgroup analysis for patients with an age cut-off of 60 years (irrespective of intervention group) showed overall mild toxicity without statistically significant differences between patients > 60 years and ≤60 years of age. The authors concluded that HD-MTX chemotherapy is a safe treatment option across different age groups in newly diagnosed PCNSL patients with adequate renal function [49].

Roth et al. assessed the outcomes of older patients (≥70 years) treated within the G-PCNSL-SG-1 trial irrespective of their allocation to the two treatment arms. Overall, older patients (≥70 years) showed significantly lower remission rates and inferior OS and PFS rates than younger patients (<70 years). When analyzing the response to HD-MTX-based chemotherapy, PFS in patients with CR was lower in older patients when compared to younger patients (mean PFS: 16.1 months in older patients (≥70 years) vs. 35.0 months in younger patients (<70 years)) [24]. Toxicity was age-independent except for a higher rate of grade 3 and 4 leukopenia in older patients (≥70 years).

##### RCTs Comparing Chemotherapy with WBRT vs. WBRT Alone (N = 1)

Mead et al. reported survival rates measured between randomization and date of death or the date “last seen alive” as primary outcome. After an earlier study closure due to poor recruitment, no statistically significant differences regarding OS were observed between the two treatment groups. On univariate log-rank analysis and multivariate Cox analysis, patient age and neurologic PS were of prognostic significance for survival irrespective of the two treatment groups. Older patients (≥60 years) with a bad pre-WBRT neurologic status had an inferior two-year OS rate of 18% (range 0–40%; 95% CI not provided) compared to younger patients (<60 years) with a good pre-WBRT neurologic status (2–year OS rate 59%, range 37–80%; 95% CI not provided) [50].

##### RCTs Evaluating HCT-ASCT (N = 2, Results Not Applicable for Elderly or Pending)

The IELSG32 phase II RCT reported two-year PFS as primary outcome: with a two-year PFS of 80% (95% CI 70–90) for 59 patients with consolidating WBRT and 69% (95% CI 59–79) for 59 patients with consolidating HCT-ASCT; no significant differences were observed between the treatment groups. Importantly, exploratory analyses showed similar survival outcomes between different age-groups: 18–59 years, 60–64 years and 65–70 (ECOG PS ≤ 2) years [32]. Nevertheless, infective complications were observed more frequently in older patients (>60 years) [51]. The ongoing MATRix/IELSG43 trial defined PFS as primary outcome. Further outcomes are CR rate, duration of response, OS, QoL, toxicity and neurotoxicity defined according to Mini-Mental State Examination, QoL and neuro-psychological battery. As follow-up is still ongoing, results are pending.

### 3.2. Prospective Non-Randomized Studies

#### 3.2.1. Key Characteristics of Prospective Non-Randomized Studies

Study key data: The key characteristics of the 26 prospective studies (25 completed, 1 ongoing) are displayed in Table 3. Two of these studies reported a control group. The remaining studies were single arm studies. The studies were published between 1992 and 2020. Only eight out of 26 studies were specifically designed for elderly patients with an age cut-off of 60 years or older [31,55,56,57,58,59,60,61]. The remaining studies included patients with a wider age range and also provided individual patient data or subgroup data for elderly patients. Nineteen studies included patients without providing an upper age limit.

Definition of patient population: While 22 studies investigated treatment in newly diagnosed PCNSL patients, four studies included patients with refractory or relapsed disease [62,63,64,65]. Adequate renal function and heterogeneous ECOG PS were frequently applied inclusion criteria for studies with HD-MTX treatment. The MARiTA pilot study (which investigated age-adapted HCT-ASCT treatment) only included patients with an ECOG PS ≤ 2 and a Cumulative Rating Illness Score Geriatrics (CIRS-G) of ≤6 (only considering symptoms not related to PCNSL) [60]. In the ongoing phase II MARTA study (which is also investigating HCT-ASCT) patients with an ECOG PS ≤ 2 are eligible [31].

Treatment protocols: Most studies focused on newly diagnosed PCNSL patients used first-line treatment protocols consisting of combined radio-chemo(immuno)therapy (8 studies) or chemo(immuno)therapy alone (14 studies). For induction treatment of newly diagnosed elderly PCNSL patients HD-MTX-based chemotherapy was applied in almost all studies (20 studies). Of note, the phase II PRIMAIN study has been the largest prospective study specifically designed for elderly PCNSL patients. This study established the combination of rituximab, HD-MTX and procarbazine as a promising treatment regimen in elderly patients [57]. Age-adapted HCT-ASCT as intensive consolidation treatment for elderly patients was investigated in one pilot study including 14 patients [60]. Furthermore, a study protocol of an ongoing phase II study investigating HCT-ASCT was identified. This study only recently completed recruitment of 51 participants [31].

In the relapse/refractory patient setting, novel agents like ibrutinib and temsirolimus as well as the alkylating agent temozolomide were investigated.

#### 3.2.2. Overall Findings of Prospective Non-Randomized Studies

Efficacy outcomes such as PFS, OS and remission rates and safety outcomes (particularly toxicity parameters) were commonly assessed. However, neurotoxicity assessment data in elderly patients were only reported in four studies investigating combination chemotherapy protocols with [66] or without radiotherapy [67], with intraventricular chemotherapy [68] and with consolidating HCT-ASCT [60]. In the study by Bessel et al. (using systemic chemotherapy combined with WBRT), severe (neuro-)toxicity was increased in older PCNSL patients (≥60 years) compared to younger patients (<60 years). Only 1/12 (8%) of younger patients (<60 years at diagnosis) showed mild cognitive dysfunction after standard dose WBRT whereas 6/10 (60%) of older patients ≥ 60 years developed dementia [69]. Overall, survival outcomes in elderly PCNSL patients were poor with encouraging but limited data for treatment protocols comprising maintenance treatment [56,59,70] or HCT-ASCT [60]. With one- and two-year PFS rates of 46.3% (95% CI 38.8–55.8) and 37.3% (28.0–46.6) and respective OS rates after one- and two-years of 56.7% (95% CI 47.2–66.1) and 47% (95% CI 37.3–56.7), the PRIMAIN regimen showed positive results but was also associated with significant toxicities. The treatment related mortality rate was 8.4% whereas 81.3% of the patients experienced grade 3 or 4 toxicities. The most frequent reported toxicities were leukopenia (55.1%), infections (35.5%), and anemia (32.7%) [57]. With two-year PFS and OS rates of 92.9% (95% CI 80.3–100) and 92.3% (95% CI 78.9–100), prospective data of the bicentric MARiTA study show promising results of the proposed age-adapted HCT-ASCT approach in this population of elderly PCNSL patients. No treatment related mortality was observed and infective complications were similar to the ones reported in the PRIMAIN study [60]. The subsequent phase II MARTA trial recently completed recruitment of 51 patients but follow-up is still ongoing and efficacy and toxicity outcomes are pending [31].

### 3.3. Retrospective Studies

#### 3.3.1. Key Characteristics of Retrospective Studies

Study key data: The key characteristics of the 24 identified retrospective studies are displayed in Appendix A. The studies were published between 1994 and 2020. Nearly half of the studies (N = 13) focused on elderly PCNSL patients. The remaining studies included a wider age range (from 12 years with no upper age limit). These studies, however, provided individual patient data or subgroup analyses for elderly patients. The study by Welch et al. reported outcomes of the oldest PCNSL patient population (≥80) [80].

Definition of patient population: Patient populations of the reported retrospective studies are heterogeneous. Only four studies included more than 100 elderly PCNSL patients (range 133–717 patients) [17,81,82,83]. The remaining studies included between 11 and 90 participants. Treatment was either based on physician’s choice and/or or performance status and age. Median age of PCNSL patients (≥65 years) undergoing HCT-ASCT was 68.5 years with a KPS of 80% [30].

Treatment protocols: Overall, first-line treatment protocols mainly consisted of MTX-based chemotherapy protocols. The largest series by Houillier et al. reported data of 717 PCNSL patients > 60 years being treated with various chemotherapy protocols. Interestingly, HD-MTX was administered even in 84% of the oldest patients aged over 80 years. However, less than half of those patients received HD-MTX doses ≥ 3 g/m^2^. Furthermore, only 2% of those patients aged > 60 years received HCT-ASCT consolidation [17]. Detailed information regarding consolidation treatment with HCT- ASCT in elderly patients was reported in only two studies [30,84]. Kassam et al. reported data for 70 patients undergoing HCT-ASCT in first response after HD-MTX containing induction treatment [84]. Schorb et al. reported data of overall 52 patients who underwent thiotepa-based HCT-ASCT with 28.8% of them receiving HCT-ASCT as first-line treatment and 71.2% of them as second or subsequent line [30].

#### 3.3.2. Key Outcomes of Retrospective Studies Investigating HCT-ASCT

Reported outcomes for elderly PCNSL patients undergoing HCT-ASCT as first-line consolidation treatment are encouraging: A European retrospective study authored by Schorb et al. investigating HCT-ASCT in elderly PCNSL patients (≥65 years) reported an overall response rate of 86.5% and a two-year PFS rate of 62% (95% CI 48.4–96) (N = 15) [30]. Another retrospective multicenter study from the UK reported by Kassam et al. investigated HCT-ASCT as first-line treatment. The study included 70 patients (age range 27–74) and concluded that age (≥5 years) was not a risk factor for treatment-related death. All 4 (from 23) patients who died in this age group ≥ 65 years) received a total thiotepa dose of 20 mg/kg. Other outcomes for elderly patients were not reported in this study [84].

## 4. Discussion

In total we identified five completed [32,42,43,45,49,50] and one ongoing RCT [29], 26 prospective studies (two with control [67,69], 24 single arm studies [31,55,56,57,58,59,60,61,62,63,64,65,66,68,70,71,72,73,74,75,76,77,78,79] and 24 retrospective studies (with or without control) [17,30,37,45,80,81,82,83,84,85,86,87,88,89,90,91,92,93,94,95,96,97,98,99] investigating different approaches in immunocompetent PCNSL patients. However, only one completed prospective pilot trial [60], one ongoing prospective phase II trial [31], and two retrospective studies [30,84] specifically investigated intensified consolidation treatment with HCT-ASCT in elderly PCNSL patients. Not only with regard to this intensive therapy regime, but in general, the data on therapy for the elderly patients with PCNSL is very limited as elderly PCNSL patients are under-represented in clinical trials. Comorbidities, poor baseline PS and potential drug toxicity are considered major issues in treating patients within this age group [23]. Elderly PCNSL patients frequently fail to receive optimal treatment owing to the lack of well-established treatment standards and geriatric assessment tools to guide treatment intensification. Recent treatment recommendations suggest that MTX should be aimed at the maximal tolerated dose for elderly patients and that dose reductions are likely to impact treatment outcomes [23]. Thus, the identification of optimal therapy and adequate dose intensity are very important factors in treating elderly PCNSL patients.

High-dose chemotherapy followed by ASCT is known to be a highly effective treatment strategy for non-Hodgkin lymphomas [100]. The rationale for HCT-ACST in PCNSL is the delivery of blood-brain-barrier penetrating agents in several-fold higher concentrations, which cannot be achieved with conventionally dosed therapy [78,101]. During the past years, 2 RCTs [32,33] have established HCT-ASCT in PCNSL patients up to the age of 65 to 70 years as a widely used treatment approach in younger PCNSL patients [30,60,102]. The RCT by Ferreri et al. included patients up to the age of 65 years or between 65 and 70 years in case of a good PS (ECOG PS ≤ 2) and provided subgroup analyses for those 2 age groups. Exploratory analyses showed that outcome for elderly patients (65–70 years with an ECOG PS of ≤2) had similar survival outcomes as younger patients [32].

Nevertheless, infective complications were observed more frequently in patients aged older than 60 years [51], underlying the important role of supportive care (including antiinfective prophylaxis) in this vulnerable subgroup of patients. This will be addressed in our randomized phase III trial by the addition of pre-specified anti-infective prophylaxis measures and a pre-phase treatment with rituximab and HD-MTX with the aim to reduce infectious complications. The ongoing randomized phase III MATRix/IELSG43 trial has recently completed recruitment; results will further define the role of HCT-ASCT in PCNSL patients up to the age of 70 years. 

The PRECIS trial by Houillier et al. is another phase II trial that compared WBRT to HCT-ASCT as consolidative strategies in PCNSL. The trial, however, included younger patients up to age 60 only (therefore it is not included in the current scoping review). Patients received rituximab, etoposide, carmustine, prednisone and cytarabine as induction therapy. Those with response received either WBRT or HCT-ASCT. Based on this results and various single arm studies [34,35,36,38] HCT-ASCT has been established as a widely used treatment approach in PCNSL patients up to the age of 65 to 70 years. These findings are supported by the results of a systemic review and meta-analysis including 43 studies and reporting outcome data of mainly thiotepa-based HCT-ASCT as consolidating or salvage treatment in PCNSL patients with a median age range between 42 and 68.5 years [102]. Alnahhas et al. reported an overall response rate of 94% after consolidative HCT-ASCT with respective two-year OS and PFS rates of 86% and 70%.

In contrast to the improvement of outcomes in younger patients, treatment strategies for elderly PCNSL patients are slow to progress and intensified treatment protocols are applied only in a small subgroup of patients.

Similar to that of those with systemic lymphoma entities, geriatric assessment could be an important tool to guide treatment intensity in elderly PCNSL patients [103]. However, to date, supporting evidence is scarce and there is only one retrospective study that investigated the impact of comorbidities on treatment feasibility and outcome measures in elderly PCNSL patients [85]. Farhi et al. investigated three comorbidity scores, the CCI, the CIRS-G and the G8. The authors found an association between a high CIRS-G score and shorter PFS and OS in univariate analysis, but these findings could not be confirmed in multivariate analysis [85]. The evaluation of applicability and utility of geriatric assessment tools should clearly be incorporated in future prospective trials, and will be also part of our planned randomized phase III PRIMA-CNS trial.

Interestingly, despite the rarity of the disease and the major therapeutic challenges, recruitment was successfully completed in all prospective studies specifically designed for elderly patients. Only one study with an age-adjusted chemotherapy approach for patients aged 31–75 years was terminated prematurely due to a high treatment related mortality rate of 16.7% [75].

The only randomized trial specifically designed for elderly PCNSL patients reported an improved efficacy without significant differences in toxicity in the more intensive chemotherapy arm comprising MTX, procarbazine, vincristine and AraC when compared to MTX and temozolomide alone [43] suggesting that this treatment approach in eligible elderly PCNSL patients may be promising.

In summary, despite encouraging data regarding the use of intensive therapy strategies even in elderly PCNSL patients, none of the reported randomized trials in PCNSL investigated an intensive treatment approach comprising HCT-ASCT in elderly PCNSL patients. Importantly, searching in different trial registers also did not reveal any ongoing randomized trial investigating this specific research question. Thus, there is an urgent need to perform a RCT addressing this treatment approach to provide elderly PCNSL with the best option currently available.

## 5. Conclusions

The results of our literature review reveal a shortage of studies that evaluate intensified chemotherapy protocols in elderly PCNSL patients. The mapping of the published literature revealed a heterogeneous study pool with regard to sample sizes, treatment protocols and toxicity data. The limitations and heterogeneity of the existing body of evidence reduce its utility for informing clinical treatment choices. Notably, clinical data regarding HCT-ASCT treatment in the elderly is scarce. Although it has been shown that HCT-ASCT is probably effective, this treatment approach has never been investigated within a well conducted RCT including a wide range of elderly patients. A thoroughly planned RCT comparing HCT-ASCT with the current treatment standard (of conventional combination chemoimmunotherapy) is, therefore, of great clinical importance to provide older PCNSL patients with the most effective treatment option.

## Figures and Tables

**Figure 1 cancers-13-04268-f001:**
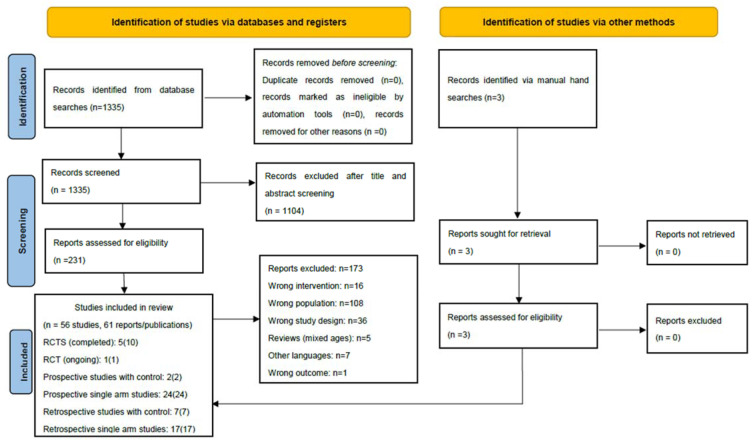
Results of the bibliographic literature searches and study 198 selection (PRISMA 2020 flow diagram). RCT: randomized controlled trial. From: [42] For more information, visit: http://www.prisma-statement.org/, accessed on 11 August 2021.

**Table 1 cancers-13-04268-t001:** Key characteristics of randomized controlled trials.

Study	Study Design (N Centres)	Country/ Recruitment	Definition of Patient Population	Study Arms	Chemotherapy	Daily Dose	Treatment Duration/ Cycle	Treatment Scheme	Additional Therapies	N Randomized	Age Years (Range/SD)	Follow-up Months
RCTs comparing different types of chemotherapy (N = 2, corresponding to 2 publications)												
Bromberg 2019 [44]	Phase III RCT (23)	The Netherlands, Australia, and New Zealand (2010–2016)	Immunocompetent patients with neuroimaging or histologically confirmed newly diagnosed PCNSL, aged 18–70 y and with ECOG PS 0–3 (Subgroup analysis for patients > 60 y)	I	Rituximab	375 mg/m^2^	2–4 days	2 × 28-day cycles	WBRT (only pts. aged ≤ 60)	100	Median 61 (55–67)	Median 32.9 (IQR 24–52)
					MTX	3 g/m^2^	2 days					
					Carmustine	100 mg/m^2^	1 day					
					Teniposide	100 mg/m^2^	2 days					
					Prednisone	60 mg/m^2^	6 days					
					AraC (consolidation)	2 × 2 g/m^2^	2 days					
				C	MTX	3 g/m^2^	2 days	2 × 28-day cycles	WBRT (only pts. aged ≤ 60)	100	Median 61 (56–66)	
					Carmustine	100 mg/m^2^	1 day					
					Teniposide	100 mg/m^2^	2 days					
					Prednisone	60 mg/m^2^	6 days					
					AraC (consolidation)							
Omuro 2015 [43]	Phase II RCT (13)	France (2007–2010)	Immunocompetent patients with neuroimaging and histologically confirmed newly diagnosed PCNSL, aged ≥ 60 and Karnofsky PS ≥ 40	I	MTX	3.5 g/m^2^	2 days	3 × 28-day cycles	/	48	Median 73 (60–85)	Median 32 (IQR 26–36)
					Temozolomide	150 mg/m^2^	5 days					
				C	MTX	3.5 g/m^2^	2 days	3 × 28-day cycles	/	47	Median 72 (60–84)	
					Procarbazine	100 mg/m^2^.	8 days					
					Vincristine	1.4 mg/m^2^.	2 days					
					AraC (consolidation).	3 g/m^2^	2 days (consolidation).					
RCT comparing chemotherapy with WBRT vs. chemotherapy alone (N = 1, corresponding to 5 publications)												
Herrlinger 2017 [48], Korfel 2015 [47], Thiel 2010 [46], Jahnke 2005 [49], Roth 2012 [24] ^°^ (G-PCNSL-SG-1)	Phase III RCT (75)	Germany (2000–2009)	Immunocompetent adult patients with newly diagnosed, histologically confirmed PCNSL (Subgroup analysis for patients > 60 y).	I	HD-MTX Ifosfamide	4 g/m^2^ 1.5 g/m^2^	3 days 3 days	6 × 14-day cycles	WBRT *	273	Median 62 (10.8)	Max: 81.2 **
				C					No WBRT	278	Median 61 (11.6)	
RCTs comparing chemotherapy with WBRT vs. WBRT alone (N = 1, corresponding to 1 publication)												
Mead 2000 [50] ^§^	Phase II RCT (12)	UK (1988–1995)	Previously untreated, immunocompetent adult patients with pathologically proven PCNSL. (Subgroup analysis for patients > 60 y).	I	Cyclophosphamide	750 mg/m^2^	NA	6 × 21-day cycles	WBRT	38	>60 yrs (n = 17)	Median 60 (12–108)
					Doxorubicin	50 mg/m^2^	NA					
					Vincristine	1.4 mg/m^2^	NA					
					Prednisone	20 mg	5 days					
				C	WBRT (No previous chemotherapy)					15	>60 yrs (n = 3)	
RCT evaluating HCT-ASCT (N = 2, corresponding to 3 publication)												
Ferreri 2016 [51], Ferreri 2017 [32] (IELSG32 trial) ^#^	Phase II RCT (53)	Italy, UK, Germany, Switzerland, Denmark (2010–2014)	Previously untreated, immunocompetent adult patients aged 18–70 y, with histologically proven PCNS. Responder were eligible for the second randomization with HCT-ASCT ^#^. (Subgroup analysis for patients > 60 y)	Ia	MTX AraC	3.5 g/m^2^ 2 g/m^2^	1 day 2 days	4 × 21 day cycles	Randomized after induction therapy to HCT-ASCT (N = 118) (I) or WBRT (C)	75	Median 58 (50–64)	Median 40 (IQR 32–49)
				Ib	MTX AraC Rituximab	3.5 g/m^2^ 2 g/m^2^ 375 mg/m^2^	1 day 2 days 2 days			74	Median 57 (53–63)	
				Ic	MTX AraC Rituximab Thiotepa	3.5 g/m^2^ 2 g/m^2^ 375 mg/m^2^ 30 mg/m^2^	1 day 2 days 2 days 1 day			78	Median 57 (53–62)	
Schorb 2016 [29] (Protocol- MATRix trial)	Phase III RCT ongoing (35) ^##^	Germany, Italy, Switzerland, Denmark, Norway (2014–2019)	Immunocompetent patients with newly-diagnosed primary central nervous system B-cell lymphoma aged 18–65 y irrespective of ECOG PS or 66–70 y (ECOG PS ≤ 2)	I	Rituximab MTX AraC Thiotepa	375 mg/m^2^ 3.5 g/m^2^ 2 × 2 g/m^2^ 30 mg/m^2^	2 days 1 day 2 days 1 day	2 × 3 week cycles	Rituximab Dexamethasone Etoposide Ifosfamide Carboplatin	220 (planned)	NA	24 months (planned)
				C					Carmustine–thiotepa conditioned HCT-ASCT			

AraC: cytarabine; ASCT: autologous stem cell transplantation; CR: complete response; ECOG PS: eastern cooperative oncology study group performance status; HCT: high-dose chemotherapy; HD-MTX: high-dose methotrexate; IQR: interquartile range; NA: not applicable; PS: performance status; WBRT: whole brain radiotherapy; Y: year; * For non-CR patients see original study. ** Different FU times (same RCT). ^°^ Post-hoc analysis of the G-PCNSL-SG-1 Trial: the study population of the G-PCNSL-SG-1 trial was divided arbitrarily into two groups with a cut-off of 70 years or older; ^§^ Patients were randomized to WBRT followed by CHOP chemotherapy or WBRT alone. ^#^ This RCT applied a two-step randomization: in the first step patients were randomized to three different combinations of chemotherapy treatment (N = 227), and then patients who showed a response or stable disease after induction treatment were randomized between WBRT and carmustine–thiotepa conditioned HCT-ASCT (N = 118). ^##^ International Extranodal Lymphoma Study Group (IELSG) will participate in the study and recruit patients too.

**Table 2 cancers-13-04268-t002:** Outcomes considered and design aspects in the randomized trials including protocol.

Study	Reported Outcome	Definition/Measure	Result
RCTs comparing different types of chemotherapy (N = 2, corresponding to 2 publications)			
Bromberg 2019 [44]	EFS (primary outcome)	Event defined as the absence of CR or unconfirmed CR at the end of protocol treatment, or relapse or death after previous CR or unconfirmed CR. CR defined as no residual gadolinium enhancement and no steroid use; unconfirmed CR as either no residual gadolinium enhancement but steroid use, or a small residual gadolinium enhancement probably related to biopsy or haemorrhage; and partial response as a more than 50% decrease in size of the contrast-enhancing tumour.	Recruitment over 6 years, patients total: N = 200, patients randomized: N = 199, patients treated per protocol: N = 69 Intention to treat analyses results: Median EFS: 10.8 months (MBVP) vs. 14.9 months (R-MBVP) 1-year EFS: 49% (MBVP) vs. 52% (R-MBVP) Subgroup analyses: Median EFS patients ≤ 60 years: 19.7 months (MBVP) vs. 59.5 months (R-MBVP) Median EFS patients 61–70 years: 8.3 months (MBVP) vs. 4.2 months (R-MBVP)
	PFS	Time from the date of registration to disease progression or death, whichever came first.	Intention to treat: 1-year PFS: 58% (MBVP) vs. 65% (R-MBVP) Subgroup analyses: Median PFS patients ≤ 60 years: 26.3 months (MBVP) vs. 59.9 months (R-MBVP) Median PFS patients 61–70 years: 19.5 months (MBVP) vs. 14.6 months (R-MBVP)
	Proportion of patients achieving a response	Proportion of patients achieving a response after induction chemotherapy.	Intention to treat: Response after induction: 86% (both intervention groups) Subgroup analyses: Response after induction patients ≤/> 60 years: no difference
	OS	Time from the date of registration to death.	Intention to treat: 1-year OS: 79% (both groups) Subgroup analyses: Median OS patients ≤ 60 years: 56.7 months (MBVP) vs. not reached (R-MBVP) Median OS patients 61–70 years: 49.2 months (MBVP) vs. 34.9 months (R-MBVP)
	Toxicity	Defined according to Common Terminology Criteria for Adverse Events (CTCAE), version 4.	Life-threatening or fatal serious adverse events: 12% (MBVP) vs. 10% (R-MBVP) TRM: 5% (MBVP) vs. 3% (R-MBVP) Subgroup analyses: Toxicity patients ≤/> 60 years: not reported
Omuro 2015 [43]	1-year PFS (primary outcome)	Progression determined by local investigators and defined as time to date of progression or death.	Recruitment over 7 years Patients total: N = 98 Patients randomized: N = 95 Intention to treat: 1-year PFS: 36% (both groups)
	OS	Time in months.	Intention to treat: Median OS: 31 months (MTX, procarbazine, vincristine, AraC) vs. 14 months (MTX, temozolomide) two-year OS: 58% (MTX, procarbazine, vincristine, AraC) vs. 39% (MTX, temozolomide)
	Toxicity	Defined according to CTCAE, version 3.	All grade 3/4 toxicities: 72% (MTX, procarbazine, vincristine, AraC) vs. 71% (MTX, temozolomide) TRM: 6% (MTX, procarbazine, vincristine, AraC) vs. 10% (MTX, temozolomide)
	Objective response	Defined according to the IPCG Response Criteria [18].	Intention to treat: ORR: 82% (MTX, procarbazine, vincristine, AraC) vs. 71% (MTX, temozolomide)
	QoL	Assessed with EORTC QLQ-C30/BN20 [52].	Substantial impairment at baseline, improvement over time with no differences between treatment groups.
	Neuropsychological evaluation	Assessed by a neuropsychologist, using global cognitive function (Mini-Mental State Examination and Mattis Dementia Rating Scale), memory (Grober and Buschke Verbal Testing), attention (Trail Making Test A and B), activities of daily living (Derouesne scale), and psychoaff ective status (Marin’s Apathy scale and Cummings’ Neuropsychiatric Inventory).	Substantial impairment at baseline, significant improvement over time in most domains with no differences between treatment groups.
RCT comparing chemotherapy with WBRT vs. chemotherapy alone (N = 1, corresponding to 5 publications)			
Korfel 2015 [47] Thiel 2010 [46] (G-PCNSL-SG-1)	OS (primary outcome)	Time until death.	Recruitment over 9 years, patients randomized: N = 551, patients treated per protocol: N = 318 Per protocol: Median OS 32.4 months (with WBRT) vs. 37.1 months (without WBRT) Subgroup analyses: Median OS patients < 60 years (irrespective of intervention): 41.7 months Median OS patients ≥ 60 years (irrespective of intervention): 24.1 months
	Rate of CR	Response was assessed by MRI or CT, and slit-lamp examination in patients with CSF or ocular involvement at 10–14 days after the third and 6th doses of MTX. CR was defined as a complete resolution of contrast-enhancing lesions on MRI or CT, and, in patients with CSF or ocular involvement at baseline, a disappearance of lymphoma cells from these sites.	Rate of CR (irrespective of intervention): 35% Rate of CR patients < 60 years (irrespective of intervention): 38% Rate of CR patients ≥ 60 years (irrespective of intervention): 33%
	PFS	Time until first progression or death.	Per protocol: Median PFS: 18.3 months (with WBRT) vs. 11.9 months (without WBRT)
	Toxic effects	According to WHO’s 1996 classification.	Grade 3 and 4 infections: 27% Subgroup analyses: Grade 3/4 infections patients < 60 years (irrespective of intervention): 18% Grade 3/4 infections patients ≥ 60 years (irrespective of intervention): 32%
	Delayed neurotoxicity	Assessed by clinical examination, and by white matter changes or brain atrophy on MRI or CT.	Radiologic confirmed delayed neurotoxicity after a median FU of 51.4 months: 71% (with WBRT) vs. 46% (without WBRT)
	Multivariate analysis	Various variables analyzed.	
Herrlinger 2017 [48] (G-PCNSL-SG-1)	QoL	Determined using the EORTC self-reporting questionnaires EORTC-QLQ-C30 and EORTCQLQ-BN20.	Negative influence of early WBRT on QoL parameters and MMSE scores.
Roth 2012 [24] (G-PCNSL-SG-1)	OS (primary outcome)	Time until death.	Subgroup analyses patients < 70 years vs. ≥ 70 years: Per protocol: Patients total: N = 126 Median OS patients < 70 years (irrespective of intervention): 26.2 months Median OS patients ≥ 70 years (irrespective of intervention): 12.5 months
	PFS	Time until first progression or death.	Per protocol: Median PFS patients < 70 years (irrespective of intervention): 7.7 months Median PFS patients ≥ 70 years (irrespective of intervention): 4.0 months
	Multivariate analysis	Various variables analyzed.	
	Toxic effects	According to WHO’s 1996 classification.	Hematologic and non-hematologic toxicity similar in patients </≥ 70 years except for grade 3/4 leukopenia (irrespective of intervention): Grade 3/4 leukopenia patients < 70 years: 21% Grade 3/4 leukopenia patients ≥ 70 years: 34%
Jahnke 2005 [49] (G-PCNSL-SG-1)	Toxic effects	According to WHO’s 1996 classification.	Subgroup analyses patients ≤ 60 years vs. > 60 years and > 70 years (irrespective of intervention): No significant differences in severity and frequency of hematological and non-hematological toxicities. Grade 3/4 infections patients ≤ 60 years: 14% Grade 3/4 infections patients > 60 years: 13% Grade 3/4 infections patients > 70 years: 14% No acute MTX-related neurotoxicity.
	Late neurotoxicity	Defined as a dementia syndrome in absence of cerebral lymphoma manifestations.	All patients (irrespective of intervention): 19.5% of patients with radiologic signs of leukencephalopathy and 7.1% with clinical evidence of late neurotoxicity.
RCTs comparing chemotherapy with WBRT vs. WBRT alone (N = 1, corresponding to 1 publication)			
Mead 2000 [50]	Survival rate (primary outcome)	Measured from the date of randomization to the date of death or the date last seen alive.	Recruitment over 7 years Patients randomized: N = 53 Intention to treat: 1-year OS: 65% (WBRT) vs. 55% (RT-CHOP) 3-year OS: 29% (WBRT) vs. 28% (RT-CHOP)
	Failure Free Survival	Defined as the time from randomization to clinical recurrence, based on clinical evidence of disease progression or death from any cause.	1-year FFS: 59% (WBRT) vs. 42% (RT-CHOP)
	Multivariate analyses	Analyses of prognostic factors and adjusted analyses of treatment effect.	
RCT evaluating HCT-ASCT (N = 2, corresponding to 3 publication)			
Ferreri 2016 [51], Ferreri 2017 [32] (IELSG32 trial)	CR after induction chemotherapy (primary outcome in first randomization: Ferreri 2016)	CR was defined as the complete disappearance of all evidence of lymphoma; partial response was defined as a 50% or greater decrease in tumour size; progressive disease was defined as at least a 25% increase in tumour size or the appearance of any new tumour lesion; and stable disease was defined as situations that did not meet any of these criteria.	Recruitment over 4.5 years Patients randomized: N = 227 (first randomization) Patients randomized: N = 118 (second randomization) Intention to treat: CR rate after induction: 23% (MTX, AraC) vs. 30% (MTX, AraC, Rituximab) vs. 49% (MATRix)
	two-year PFS (primary outcome in second randomization: Ferreri 2017)	Estimated according to Revised Response Criteria for Malignant Lymphoma [53]. Time zero for PFS was the date of trial registration.	Intention to treat: two-year PFS: 36% (MTX, AraC) vs. 46% (MTX, AraC, Rituximab) vs. 61% (MATRix)
	Toxicity	Assessed separately for each chemotherapy course and graded according to the CTCAE, version 3.0	Grade 3/4 neutropenia/infections: 21% (MTX, AraC) vs. 14% (MTX, AraC, Rituximab) vs. 16% (MATRix) TRM: 6%
	OS	Estimated according to Revised Response Criteria for Malignant Lymphoma.	Intention to treat: two-year OS: 42% (MTX, AraC) vs. 56% (MTX, AraC, Rituximab) vs. 69% (MATRix) Exploratory subgroup analyses in patients treated with MATRix and consolidating WBRT/HCT-ASCT: Similar survival outcomes between different age groups (18–59 years, 60–64 years, 65–70 years)
	Relapse rates	Not defined.	
	Neurotoxicity	The effect of treatment on neurocognitive functions was assessed by MMSE and a panel of neuro psychological tests presently used by the IPCG [54], which were done before and after induction and consolidation treatments and every 6 months afterwards.	Significant impairment in some attention and executive functions in patients treated with WBRT. Significant improvement in attention and executive functions, memory, and quality-of-life figures in patients treated with HCT-ASCT.
Schorb 2016 [29] (Protocol-MATRix Trial)	PFS (primary outcome)	Time from the date of randomization to the date of lymphoma progression, relapse or death from any cause with possible censoring at the date of last visit of follow-up.	NA (ongoing trial, results pending)
	CR rate	On day 60 after randomization.	
	Duration of response	Time from CR, unconfirmed CR or PR until relapse, death or last follow-up visit.	
	OS	Time from randomization until death of any cause up to 24 months after end of treatment.	
	QoL	Defined according to EORTC QLQ-C30.	
	Adverse events, toxicity	Defined according to the CTCAE, version 4.0.	
	Neurotoxicity	Defined according to Mini-Mental State Examination (MMSE), EORTC QLQ-BN20 and neuro-psychological battery.	

AraC: cytarabine; ASCT: autologous stem cell transplantation; CHOP: cyclophosphamide, doxorubicin, vincristine, prednisone; CR: complete remission; CSF: cerebrospinal fluid; CTCAE: common terminology criteria for adverse events; EFS: event free survival; FFS: failure free survival; FU: follow up; HCT: high-dose chemotherapy; IPCG: international PCNSL collaborative study group; MATRix: methotrexate, cytarabine, thiotepa, rituximab; MBVP: methotrexate, carmustine, teniposide, prednisone; MTX: methotrexate; NA: not applicable; ORR: overall response rate; OS: overall survival; PFS: progression free survival, PR: partial remission; RT: radiotherapy; R-MBVP: rituximab, methotrexate, carmustine, teniposide, prednisone QoL: quality of life; WBRT: whohle brain radiotherapy.

**Table 3 cancers-13-04268-t003:** Key characteristics of prospective, non-randomized studies including efficacy and toxicity outcomes.

Scheme 2.	Title	Population	Treatment	Result
Prospective studies with control group evaluating chemotherapy + WBRT (first-line therapy) (N = 2)				
Bessell 2002 [69]	Importance of WBRT in the outcome of patients with PCNSL: an analysis of the CHOD/BVAM regimen followed by two different WBRT treatments	Adult patients (range 21–70) newly diagnosed with PCNSL. (Age subgroup analyses available).	CHOD/BVAM with WBRT (dose comparisons).	Patients total: N = 57 (31/26) Median OS (irrespective of intervention): 40 months 3-year OS (irrespective of intervention): 55% 5-year OS (irrespective of intervention): 36%. Toxicity: 5 patients (8.8%) died during chemotherapy without evidence of lymphoma (irrespective of intervention). Subgroup analyses: 3-years OS patients < 60 years who achieved CR after treatment: 92% (WBRT 45 Gy) vs. 60% (WBRT 30.6 Gy) Neurotoxicity patients < 60 years: mild cognitive dysfuntion in 8% of patients (WBRT 45 Gy) versus 0% (WBRT 30.6 Gy) Neurotoxicity patients ≥ 60 years: dementia in 60% of patients (WBRT 45 Gy) versus 0% (WBRT 30.6 Gy)
Ichikawa 2014 [67]	Reduced neurotoxicity with combined treatment of HD-MTX, cyclophosphamide, doxorubicin, vincristine and prednisolone (M-CHOP) and deferred WBRT for PCNSL	Newly diagnosed adult immunocompetent patients with PCNSL. (Age subgroup analyses available).	HD-MTX, cyclophosphamide, doxorubicin, vincristine and prednisolone (M-CHOP), with or without WBRT.	Patients total: N = 24 (9/15) Median OS: 33 months (M-CHOP + WBRT) vs. 32 months (M-CHOP alone) Toxicity: No TRM Subgroup analyses: Median OS patients > 65 years: 14 months (M-CHOP + WBRT) vs. 32 months (M-CHOP alone) Neurotoxicity: 2 patients > 65 years treated with M-CHOP + WBRT developed neurotoxicity.
Prospective single arm studies first-line (N = 20)				
Studies evaluating chemotherapy + radiotherapy, first line (N = 6)				
Bessell 2001 [66]	CHOD/BVAM regimen plus WBRT in patients with PCNSL	Newly diagnosed PCNSL patients (age range 21–70) were entered into this phase II study between February 1990 and February 1996. None of the 31 patients had clinical evidence of human immunodeficiency virus type I infection, had undergone organ transplantation, or had any previous malignancy. (Age subgroup analyses available).	CHOD/BVAM regimen and WBRT.	Patients total: N = 31 (≥60 years: N = 8) Median OS: 38 months 3-and 5-year OS 55% and 31% Median PFS: 38 months 5-year PFS: 31% Toxicity: 2 patients died during chemotherapy without evidence of PCNSL. Neurotoxicity: Dementia probably related to treatment in 62% of patients ≥ 60 years, 4 of them died without evidence of relapse of PCNSL. Subgroup analyses: 4-year PFS patients < 60 years: 58% 4-year PFS patients ≥ 60 years: 29%
Cho 2014 [55]	Pilot study of gamma-knife surgery-incorporated systemic chemotherapy omitting WBRT for the treatment of elderly PCNSL patients with poor prognostic scores	Patients aged 65 or older, with pathologically proven PCNSL, ECOG PS of grade 3 or less, and no prior use of chemotherapy and R. (4 patients in total, individual data available).	Combination chemotherapy (thiotepa, vincristine, MTX with leucovorine rescue) and gamma-knife surgery, omitting WBRT.	Patients total: N = 4 Median OS 15.8 months Median PFS 9.5 months TRM: 0%
Nelson 1992 [70]	Non-Hodgkin’s lymphoma of the brain: can high dose, large volume radiation therapy improve survival? Report on a prospective trial by the Radiation Therapy Oncology Group (RTOG): RTOG 8315	Age ≥ 18 years, KPS ≥ 40, biopsy-proven non-Hodgkin’s lymphoma involving the parenchyma of the brain without involvement of the spinal cord by myelogram or of the cerebral spinal fluid by cytology. (Age subgroup analyses available).	WBRT and meningeal irradiation. Chemotherapy given only if disease progress observed. Chemotherapy included MTX or MTX, bleomycin, doxorubicin, cyclophosphamide, vincristine, dexamethason.	Patients total: n = 46 (<60 years: n = 14, ≥60 years: n = 27) Median OS: 54.5 months Subgroup analyses: Median OS patients < 60 years: 23.1 months Median OS patients ≥ 60 years: 7.6 months
O’Neill 1995 [71]	PCNSL: survival advantages with combined initial therapy?	Eligible for enrollment were immunocompetent patients with untreated PCNSL, aged 65 years or older. Patients were required to have a neuropathological diagnosis of PCNSL.	Chemoimmunotherapy with rituximab, MTX, procarbazine, and lomustine.	Patients total: N = 28 (≥80 years: N = 6) Median OS: 17.5 months 1-year OS: 67.9% 3-year OS: 31.1% Median PFS: 16 months Toxicity: TRM: 7% Subgroup analyses: Median OS patients < 80 years: 29 months 1-year OS patients < 80 years: 82% 3-year OS patients < 80 years: 40% Median OS patients ≥ 80 years: 4.3 months 1-year OS patients ≥ 80 years: 17%
O’Neill 1999 [72]	PCNSL: survival advantages with combined initial therapy? North Central Cancer Treatment Group (NCCTG) Study 86-72-52	All patients must have had an intracranial space-occupying lesion(s) on clinical grounds, confirmed by neuroimaging (CT or MRI). The lesion must have been surgically sampled, either by biopsy or attempted resection, and pathologically confirmed as PCNSL. Median age of the eligible treated patients was 60 years, with ages ranging from 24 to 75 years (Age subgroup analyses available).	Treatment consisted of cyclophosphamide, adriamycin, vincristine, prednisone and WBRT. Followed by HD-AraC.	Patients total: N = 53 (≤60 years: N = 27, >60 years: N = 26, <70 years: N = 43, ≥70 years: N = 10) Median OS: 42 weeks 1-year PFS: 31% TRM: 3.7% Subgroup analyses: Median OS patients ≤ 60 years: 48 weeks Median OS patients > 60 years: 37 weeks Median OS patients < 70 years: 48 weeks Median OS patients ≥ 70 years: 25 weeks 1-year PFS patients ≤ 60 years: 38% 1-year PFS patients > 60 years: 23% 1-year PFS patients < 70 years: 32% 1-year PFS patients ≥ 70 years: 21%
Shibamoto 1999 [73]	Systemic chemotherapy with vincristine, cyclophosphamide, doxorubicin and prednisolone following WBRT for PCNSL: a phase II study	Among 36 patients with PCNSL seen at the Department of Radiology, Kyoto University Hospital between March 1981 and December 1995, 23 patients were treated with the radiation–VEPA chemotherapy protocol. There were 15 men and 8 women. Their ages ranged from 24 to 77 years, with a median age of 59 years. (Age subgroup analyses available).	Systemic chemotherapy with vincristine, cyclophosphamide, doxorubicin and prednisolone following WBRT.	Patients total: 23 (<60 years: N = 12, ≥60 years: N = 11) Median OS: 25.2 months 5-year OS: 23% Toxicity: decline in performance status unaccompanied with tumor recurrence observed in 44% of the patients. Subgroup analyses: Incidence of decline in performance status unaccompanied with tumor recurrence higer than in younger patients.
Studies evaluating chemo(immuno)therapy, first line (N = 12)				
Fritsch 2011 [56]	Chemoimmunotherapy with rituximab, HD-MTX, procarbazine, and lomustine for PCNSL in the elderly	Eligible for enrollment were immunocompetent patients with untreated PCNSL, aged 65 years or older. Patients were required to have a neuropathological diagnosis of PCNSL.	Chemoimmunotherapy with rituximab, HD-MTX, procarbazine, and lomustine.	Patients total: n = 28 (≥80 years: n = 6) Median OS: 17.5 months 1-year OS: 67.9% 3-year OS: 31.1% Median PFS: 16 months Toxicity: TRM: 7% Subgroup analyses: Median OS patients < 80 years: 29 months 1-year OS patients < 80 years: 82% 3-year OS patients < 80 years: 40% Median OS patients ≥ 80 years: 4.3 months 1-year OS patients ≥ 80 years: 17%
Fritsch 2017 [57]	HD-MTX-based chemoimmunotherapy for elderly PCNSL patients (PRIMAIN study)	Immuno-competent patients with newly diagnosed PCNSL (proven on histology) according to the World Health Organization criteria, aged 65 or older were eligible irrespective of clinical performance status.	HD-MTX-based immuno-chemotherapy.	Patients total: n = 107 Median OS 20.7 months 1-year OS: 56.7% 2-year OS: 47% Median PFS: 10.3 months 1-year PFS: 46.3% 2-year PFS: 37.3% TRM: 8.4%
Ghesquieres 2010 [74]	Long-term follow-up of an age-adapted C5R protocol followed by WBRT in 99 newly diagnosed PCNSL: a prospective multicentric phase II study of the Groupe d’Etude des Lymphomes de l’Adulte (GELA)	Patients older than 18 years with unknown cause of immunodepression and newly diagnosed PCNSL were eligible. All histological subtypes were allowed. Diagnosis was obtained by histological biopsy or cytological analysis of the CSF. (Age subgroup analyses available).	Age-adapted C5R protocol: cyclophosphamide, vincristine, prednisone, MTX, cyclophosphamide, doxorubicin, vincristine, prednisone, AraC.	Patients total: N = 99 (<61 years: N = 45, 61–70 years: N = 36, > 70 years: N = 18) Median OS: 33 months 5-year OS: 42% Median PFS: 20 months 5-year PFS: 26% Toxicity: 9% of patients died of infectious causes. Subgroup analyses: Median OS patients < 61 years: 46 months Median OS patients 61–70 years: 16 months Median OS patients >70 years: 15 months Median PFS patients < 61 years: 28 months Median PFS patients 61–70 years: 16 months Median PFS patients > 70 years: 7 months 5-year PFS patients < 61 years: 31% months 5-year PFS patients 61–70 years: 28% months 5-year PFS patients > 70 years: 11%
Goldkuhl 2002 [75]	Age-adjusted chemotherapy for PCNSL--a pilot study	Patients aged 31–75 years, with a diagnosis of PCNSL and with no evidence of systemic disease. The eligibility criteria for patients included performance status 0–2, normal renal function, no need for anti-inflamatory drugs and no immunosuppression. (Individual data available).	Patients aged < 65 recieved Carmustine, vincristine, dexamethasone, HD-MTX and HD-AraC. Patients aged > 65 recieved the same therapy without HD-MTX.	Patients total: N = 30 (>65 years: N = 13) Recruitment was stopped prematurely after 30 patients due to high TRM rate of 16.7% (<65 years: N = 2/17 (11.7%); >65 years: N = 3/13 (23%) Median OS patients < 65 years: not reached Median OS patients > 65 years: 15 months
Hoang-Xuan 2003 [58]	Chemotherapy alone as initial treatment for PCNSL in patients older than 60 years: a multicenter phase II study (26952) of the European Organization for Research and Treatment of Cancer Brain Tumor Group	Patients were considered eligible if the diagnosis of PCNSL was histologically confirmed (by brain biopsy, CSF cytology, or vitrectomy), if age was 60 years or older, and if KPS was ≥40.	The protocol consisted HD-MTX, lomustine, procarbazine, methylprednisolone, and intrathecal chemotherapy with HD-MTX and AraC.	Patients total: N = 50 Median OS: 14.3 months 1-year OS: 52% Median PFS: 10.6 months 1-year PFS: 47% Toxicity: 1 death during therapy caused by pulmonary embolism
Illerhaus 2009 [59]	HD-MTX combined with procarbazine and CCNU for PCNSL in the elderly: results of a prospective pilot and phase II study	Immunocompetent patients with untreated PCNSL or intraocular lymphoma aged 65 years or older, or those below the age of 65 otherwise unfit for our simultaneously initiated highdose chemotherapy protocol with ASCT.	HD-MTX combined with procarbazine and lomustine.	Patients total: N = 30 Median OS: 15.4 months 3- and 5-year OS: 33.3% Median PFS: 5.9 months Toxicity: TRM: 6.7%
Juergens 2010 [76]	Long-term survival with favorable cognitive outcome after chemotherapy in PCNSL	Adult patients (age range: 27–75 years) with histologically proved PCNSL. (Age subgroup analyses available).	HD-MTX, Ara-C, combined with dexamethasone, vinca-alkaloids, ifosfamide, and cyclophosphamide.	Patients total: N = 65 (patients ≤ 60 years: N = 35) Median OS: 54 months Median TTF: 21 months Subgroup analyses: Median OS patients ≤ 60 years: not reached Median OS patients > 60 years: 34 months Median TTF patients ≤ 60 years: 47 months Median TTF patients > 60 years: 7 months
Pels 2003 [68]	PCNSL: results of a prospective pilot and phase II study of systemic and intraventricular chemotherapy with deferred WBRT	Patients aged 18–75 with newly diagnosed histologically proven non Hodgkin’s lymphoma, according to the Revised European-American Lymphoma and WHO classification. Patients with lymphoma that involved sites other than the brain, meninges, CSF, or the eyes were not included. (Age subgroup analyses available).	HD-MTX and AraC based systemic therapy including dexamethasone, vinca-alkaloids, ifosfamide, and cyclophosphamide combined with intraventricular MTX, prednisolone, and ARA-C.	Patients total: N = 65 (<61 years: N = 30, >60 years: N = 35) Median OS: 50 months Toxicity: TRM: 9% Neurotoxicity: permanent cognitive dysfunction (possibly treatment-related): 3% Subgroup analyses: Median OS patients > 60 years: 34 months Median OS patients < 61 yeears: not reached TRM patients < 61 years: 6.9% TRM patients > 60 years: 12.5%
Pulczynski 2015 [77]	Successful change of treatment strategy in elderly patients with PCNSL by de-escalating induction and introducing temozolomide maintenance: results from a phase II study by the Nordic Lymphoma Group	Immunocompetent patients aged 18–75 years with newly diagnosed histologically-confirmed PCNSL. Patients pre-treated with steroids were eligible. There were no limitations for inclusion with regard to ECOG PS. (Age subgroup analyses available).	HD-MTX, AraC and maintenance with temozolomide.	Patients total: N = 66 (≤65 years: N = 39, 66–75 years: N = 27) 2-year OS: 58.7% TRM: 6% (all patients aged 64–75 years) Subgroup analyses: 2-year OS patients ≤ 65 years: 60.7% 2-year OS patients > 65 years: 55.6% 2-year PFS patients ≤ 65 years: 33.1% 2-year PFS patients > 65 years: 44%
Rubenstein 2013 [78]	Intensive chemotherapy and immunotherapy in patients with newly diagnosed PCNSL: CALGB 50202 (Alliance 50202)	Patients (age range: 12–76) with histologic confirmation of PCNSL, with central review of diagnostic specimens. Measurable disease based on gadolinium enhancement of brain or spine MRI and/or positive CSF cytology was also required. (Age subgroup analyses available).	HD-MTX, temozolomide, and rituximab with leucovorin rescue. Consolidation with etoposide, AraC.	Patients total: N = 44 Median OS: not reached 4-year OS: 65% Median PFS: 2.4 years TRM: 2.2% Subgroup analyses: Patients > 60 years experienced outcomes similar to those of younger patients.
Sung 2011 [79]	Factors influencing the response to high dose MTX-based vincristine and procarbazine combination chemotherapy for PCNSL	Immunocompetent adult patients, aged 18 or older, with newly diagnosed histologically proven PCNSL. (Age subgroup analyses available).	Modified RTOG 93-10 protocol: MTX followed by leucovorin rescue, vincristine, and procarbazine.	Total patients: N = 52 (<60 years: N = 17, ≥60 years: N = 35) Median OS: 30.5 months 2-year OS: 62.3% Toxicity: TRM: 3.8% Subgroup analyses: Median OS patients < 60 years: 32.3 months Median OS patients ≥ 60 years: 27.3 months Median PFS patients < 60 years: 21.7 months Median PFS patients ≥ 60 years: 18.7 months
Zhu 2009 [61]	HD-MTX for elderly patients with PCNSL	Consecutive patients with pathologically confirmed PCNSL involving the brain, the eyes, or both who were 70 years or older at the time of diagnosis.	Treatment included MTX monotherapy only.	Total patients: N = 31 Median OS: 37 months Median PFS: 7.1 months Toxicity: Grade 3 or 4 toxicity: 9.7%
Prospective single arm studies investigating HCT-ASCT, first-line (N = 2)				
Schorb 2020 [60]	HCT-ASCT in elderly patients with primary CNS lymphoma: a pilot study	Newly diagnosed immunocompetent elderly patients, aged 65 or older, with histologically proven PCNSL of B-cell immunophenotype, (excluding isolated primary vitreoretinal lymphoma), ECOG PS ≤ 2 and Cumulative Illness Rating Scale–Geriatric score ≤ 6.	Induction chemotherapy consisting of rituximab, HD-MTX and AraC followed by HCT-ASCT.	Patients total: N = 14 2-year OS: 92.3% 2-year PFS: 92.9% TRM: 0%
Schorb 2019 [31]	Age-adjusted HCT-ASCT in elderly and fit PCNSL patients	Patients must be immunocompetent, newly-diagnosed with histologically proven PCNSL of B-cell immunophenotype, aged 65 years or older with an ECOG PS ≤ 2, and eligible for HCT-ASCT as to the treating physician.	Induction chemotherapy consisting of rituximab, HD-MTX, and AraC. After 2 cycles of induction chemotherapy, patients achieving at least stable disease will undergo HCT-ASCT.	Patient total (planned): N = 51 (ongoing study, results pending)
Prospective single arm studies, relapsed/refractory disease (N = 4)				
Korfel 2016 [62]	Prospective Phase II Trial of Temsirolimus for Relapsed/Refractory PCNSL	Eligibility criteria included patients older than 18 years with PCNSL proven histologically or cytologically and with evidence of a relapse or progression (on MRI or in the CSF) after HDMTX-based primary chemotherapy. (Data for each patient available).	Temsirolimus with clemastine premedication.	Patients total: N = 37 (median age 70 years) Median OS 3.7 months Median PFS: 2.1 months TRM: 13.5%
LOC Study Langner-Lemercier 2016 [63]	PCNSL at first relapse/progression: characteristics, management, and outcome of 256 patients from the French LOC network	PCNSL patients were included in the study if they fulfilled the following criteria: (1) age over 18 years, (2) pathologically proven PCNSL at initial diagnosis, (3) refractory or relapsed disease after first-line therapy, and (4) clinical data on patient characteristics, treatment, and outcome available for analysis. (Age subgroup analyses available).	The choice of the treatments (initial and salvage therapy) was left to the discretion of the treating physicians.	Patients total: N = 256 (≥60 years: N = 199) Median OS: 2.2 months Median PFS: 3.5 months
LOC Study Soussain 2019 [64]	Ibrutinib monotherapy for relapse or refractory PCNSL and primary vitreoretinal lymphoma: Final analysis of the phase II ‘proof-of-concept’ iLOC study by the Lymphoma study association (LYSA) and the French oculo-cerebral lymphoma (LOC) network	Immunocompetent adult patients with relapse or refractory PCNSL or primary vitreoretinal lymphoma were eligible if they had received prior HD-MTX and had an ECOG PS < 2. (Age subgroup analyses available).	Ibrutinib until disease progression or unacceptable toxicity occurred. Additional corticosteroid treatments were allowed during the first 4 weeks of treatment.	Patients total: N = 52 (≥60 years: N = 35, <60 years: N = 17) Median OS: 4.8 months Median PFS: 19.2 months Subgroup analyses: age and Toxicity: fatal pulmonary aspergillosis in 1 patient (1.9%)
Reni 2004 [65]	Salvage chemotherapy with temozolomide in PCNSL: preliminary results of a phase II trial	Patients aged age 18–75 years with PCNSL failure after previous treatment including HD-MTX and/or RT, histological or cytological diagnosis of non-Hodgkin’s lymphoma, disease limited to the brain, presence of at least 1 bi-dimensionally measurable target lesion, negative human immunodeficient virus serology, ECOG PS < 4. (Data for each patient available).	Eligible patients received oral temozolomide.	Patients total: N = 23 (median age 60 years) Median OS: 3.5 months Median PFS: 2 months

AraC: cytarabine; ASCT: autologous stem cell transplantation; BVAM: bis-chloronitrosourea, cytosine arabinoside, methotrexate; CHOD: cyclophosphamide, doxorubicin, vincristine, and dexamethasone; CT: computed tomography; Gy: grey; HD-AraC: high-dose cytarabine; HD-MTX: High-dose methotrexate; HCT: high-dose chemotherapy; KPS: karnofsky performance status; MRI: magnetic resonance imaging; OS: overall survival; PCNSL: primary central nervous system lymphoma; PFS: progression free survival; R-GEMOX: gemcitabine, oxaliplatine, rituximab; RT: radiotherapy; TTF: time to treatment failure; TRM, treatment related mortality; WBRT: whole brain radiotherapy.

## Data Availability

Data is contained within the article or Appendix A. The primary data presented in this systematic review are available in the primary studies cited in the reference list.

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
