# Peer review of "Treatment Regimens for Immunocompetent Elderly Patients with Primary Central Nervous System Lymphoma: A Scoping Review"

_cancers, 2021, doi:10.3390/cancers13174268_

Round 1
Reviewer 1 Report
This manuscript, written by Dr. Elisabeth Schorb, with the title of “Treatment Regimens for Immunocompetent Elderly Patients with Primary Central Nervous System Lymphoma: A Scoping Review,” is a review manuscript that focused on the treatment of Primary central nervous system lymphoma (PCNSL). This manuscript performed a systematic review of PCNSL and had a special interest in immunocompetent patients of the elderly.
PCNSL is an infrequent subtype of extranodal non-hodgkin lymphoma (NHL) that involves the brain, leptomeninges, eyes, or spinal cord without evidence of systemic disease. The epidemiology and the clinical presentation depends whether the patient is immunocompromised. Aside from the importance of high-dose systemic methotrexate (MTX), there is little consensus on the optimal components of induction and consolidation therapy for newly diagnosed PCNSL, and there is variation in clinical practice.
This review is well written and makes a thorough description of the different studies. Before publishing this manuscript, the authors may find the following comments useful for improving their work.
1) The authors could improve the discussion of the clinical presentation, predisposing factors, pathologic features, and diagnosis of PCNSL in immunocompetent patients. For example, the immune microenvironment, genomic landscape, mutations, the presence of Epstein-Barr virus (EBV), IgHV status and cell-of-origin.
2) The immune system changes with age. Elderly patients are in a state of “acquired immune suppression.” If the authors agree with this statement, they could comment about PCNSL in immunocompromised patients.
3) On page 40 of 47, lines 547-550 should be deleted because these are the MDPI instructions for the discussion.
4) There is a review of this subject in UpToDate. This review also have a section for older adults. The authors may find it useful to read it and incorporate some information that may improve their manuscript.
The UpToDate sections is the following:
Treatment and prognosis of primary central nervous system lymphoma
Batchelor T. et al.
UpToDate 2021.
“Older adults — Treatment decisions in older adults with PCNSL must be individualized, taking into account not only age but also functional status and comorbidities. A comprehensive geriatric assessment may be useful in assessing comorbidity and functional status in the older adult patient, thus permitting the formulation of an appropriate, individualized treatment plan. Special considerations for the use of chemotherapy in the older adult population are discussed separately. (See "Comprehensive geriatric assessment for patients with cancer" and "Systemic chemotherapy for cancer in older adults".)
In a systematic review and meta-analysis that included over 700 patients with PCNSL diagnosed between the ages of 60 and 90 years, Karnofsky Performance Status (KPS) ≥70 was a much stronger prognostic factor for overall survival than age alone [11]. After adjusting for KPS, only age >75 years remained significantly associated with increased mortality. As in younger adults, high-dose MTX-based therapies were associated with better outcomes compared with non-MTX-based therapies, and receipt of WBRT was associated with an increased risk of neurologic side effects.
First-line therapy – For older adults with PCNSL who are candidates for chemotherapy, we suggest initial high-dose MTX-based therapy plus rituximab as used for younger patients, either with MTX alone (for those with borderline functional status) or combined with an oral alkylating agent such as temozolomide or procarbazine (algorithm 2). Although most studies have used a cutoff of 60 years to define older adults, we more commonly use 70 years in clinical practice.
Numerous small, mostly single-center prospective or retrospective studies have evaluated various induction regimens in older adults with PCNSL [11,21,74-76]. In aggregate, the available data suggest that high-dose MTX-based combination regimens may result in better response rates than high-dose MTX monotherapy, and that combination regimens containing an oral alkylating agent such as temozolomide or procarbazine may be as effective as combination regimens containing two or more intravenous agents [11].
The following studies illustrate the results of specific regimens in the treatment of older adults with PCNSL:
A randomized phase II trial of high-dose MTX, procarbazine, and vincristine plus cytarabine (MPV-A) versus high-dose MTX plus temozolomide in 95 older adults (age ≥60 years) [77]. There was a nonsignificant trend in favor of MPV-A for all outcomes, including objective response rate (82 versus 71 percent), progression-free survival (PFS; 9.5 versus 6.1 months), and overall survival (31 versus 14 months). Severe (grade 3/4) toxicities occurred in approximately 70 percent of patients in both groups. A subsequent retrospective study in older adults found that an intensified MPV-A regimen, with three monthly cycles of high-dose cytarabine instead of one, was associated with high toxicity and no clear advantage in terms of efficacy outcomes [78].
Rituximab, high-dose MTX (3 g/m2), and procarbazine (R-MP) with or without lomustine (R-MPL) has been studied in several phase II trials in older adults [74,75,79]. In the largest multicenter study, 107 older adults (age ≥65 years) were treated with R-MPL (n = 69) or R-MP (n = 38); lomustine was omitted after the first 69 patients were enrolled due to infectious complications [79]. Compared with R-MPL, R-MP was associated with slightly lower overall response rate (32 versus 38 percent) and two-year PFS (35 versus 39 percent) but similar two-year overall survival (48 versus 46 percent; median overall survival 21 months). Grade 3/4 toxicities were common in both groups (71 versus 87 percent).
Consolidation – The optimal consolidation approach for older patients who respond to induction therapy has not been established, and treatment decisions are individualized. We do not routinely treat older patients with high-dose consolidative chemotherapy, and we suggest postponing WBRT until the time of progressive disease (PD) rather than delivering it after the completion of induction chemotherapy. However, highly selected older adults may be eligible for, and benefit from, an attempt at high-dose chemotherapy consolidation.
This was illustrated by a prospective two-center trial in which 37 older adults (>65 years) with newly diagnosed PCNSL were referred and 14 patients were ultimately treated with a protocol consisting of a short induction regimen (two 21-day cycles of rituximab, high-dose MTX [3.5 g/m2], and cytarabine [2 g/m2 twice a day for two days]) followed by busulfan-based high-dose chemotherapy with autologous hematopoietic cell transplantation (HCT) rescue [80]. Overall, 13 patients completed the full protocol, and there were no treatment-related deaths. Grade 3/4 hematologic toxicities were seen in all patients. After induction therapy, there were 3 complete responses (CRs), 1 unconfirmed CR, and 10 partial responses (PRs). With a median follow-up of 41 months, one patient progressed 9 months after HCT rescue, and all other patients had ongoing CRs; two-year PFS and overall survival were both 92 percent. A larger multicenter study is planned.
Maintenance – For patients who are not selected for consolidation, an alternative maintenance approach used by some centers in older adults consists of monotherapy with either monthly high-dose MTX or a targeted agent such as lenalidomide [81,82]. (See 'Nonmyeloablative chemotherapy or targeted agents' above and 'Lenalidomide' below.)
Vitreoretinal disease — The optimal management of vitreoretinal lymphoma remains unclear, and there is little consensus among ophthalmologists [83-88]. Lacking prospective randomized trials comparing different therapies, the majority of data come from retrospective analyses.
For patients with PCNSL with vitreoretinal involvement, we suggest the administration of dedicated ocular therapy (either radiation or intravitreal chemotherapy) in addition to systemic therapy, rather than systemic therapy alone or ocular therapy alone [84,89,90]. Alternatively, dedicated ocular therapy may be reserved for those patients who do not obtain a CR with high-dose systemic chemotherapy alone.
Ocular-directed therapy has been added principally based upon the concern that it may be difficult to achieve active concentrations of chemotherapy in the eye with systemic therapy alone [90]. However, some patients with ocular involvement given systemic chemotherapy achieve sufficient concentrations of MTX in the vitreous and aqueous humor to provide a disappearance of cells in this location (ie, CR) [15]. Patients treated with intravitreal therapy in addition to systemic therapy appear to have a longer median PFS, but similar overall survival rates when compared with those who do not receive dedicated ocular therapy.
The benefit of dedicated ocular therapy was reviewed in an international retrospective series that included 221 HIV-seronegative, immunocompetent patients with PCNSL with vitreoretinal involvement [91]. Treatment information was available for 176 patients and included dedicated ocular therapy in 102 patients (ocular radiotherapy in 79 patients, intravitreal MTX in 22 patients) in addition to more global treatment of their brain lymphoma. Patients who received dedicated ocular therapy had significantly longer median PFS (19 versus 15 months) but no difference in median overall survival (31 months) when compared with those who did not receive dedicated ocular therapy. The pattern of treatment failure did not differ between these two groups. Sites of progression included the brain, eyes, brain and eyes, or systemic disease in 52, 19, 12, and 2 percent, respectively. Another multicenter retrospective series that included 78 patients with primary vitreoretinal lymphoma found a slightly lower rate of progression in brain (36 percent) and a slightly longer median overall survival (44 months); neither endpoint was influenced by type or extent of initial therapy [92].
The preferred dedicated ocular therapy remains to be determined. Either ocular radiation or intravitreal chemotherapy can be used. Ocular radiation therapy is the most commonly used dedicated ocular therapy given its ease of administration, but intravitreal MTX is an acceptable alternative.
Ocular radiation – Ocular radiation, given at a total radiation dose of 35 to 40 Gy fractionated over five weeks, has been the standard therapy for many years. Patients require treatment of both eyes and will have an improvement in their vision with vitreal clearing, but it is frequently followed by xerophthalmia and chemosis. Cataracts, corneal ulcerations, and retinal injuries are uncommon [93,94]. (See "Delayed complications of cranial irradiation", section on 'Xerophthalmia'.)
Intravitreal MTX – A prospective trial evaluated the use of intravitreal MTX (400 micrograms in a total volume of approximately 0.1 mL) twice weekly until response is achieved, followed by weekly injections for one month, and then monthly injections for one year [95]. In a follow-up study of 16 HIV-seronegative patients with intraocular B cell lymphoma, all 26 involved eyes were cleared clinically of malignant cells after a maximum of 12 intravitreal injections [96]. Side effects were common and included cataract, corneal epitheliopathy, and maculopathy; no patient had irreversible loss of vision that could be definitely attributed to the MTX injections. The intravitreous administration of rituximab in this population is under exploration.
Intravitreal rituximab – The anti-CD20 monoclonal antibody, rituximab, has become a standard component of the initial systemic therapy for patients with CD20-positive lymphomas. Small case series have suggested that intravitreal rituximab is safe and has activity against primary intraocular lymphoma when administered alone or in combination with MTX [97,98].”

Reviewer 2 Report
This is an interesting scoping review, helpful to summarize research findings in the field of treatment regimens for immunocompetent elderly patients with primary central nervous system lymphoma. Although actually there are only few studies that evaluate HCT ASCT treatment in this subgroup of elderly patients, results of the phase III RCT ongoing trial will be soon available and will probably provide at least a partial response to the authors’ question.
Reviewer 3 Report
The authors review the clinical treatment and management of primary central nervous system lymphoma in elderly patients. While the clinical trials including chemotherapies and combinations of high-dose chemotherapy followed by autologous stem cell transplantation for PCNSL have been extensively investigated in randomized controlled trials, prospective, and retrospective studies among young populations, studies using HCT-ASCT strategy to treat the elderly is scarce. The existing treatment options result in poor prognosis and efficacy. After reviewing the 6 randomized controlled trials, the authors conclude that the study pool is not conclusive due to the small sample size and heterogeneous patient populations. The limited data on therapy for elderly patients with PCNSL preclude reliable evaluation and treatment with the HCT-ASCT strategy. The authors appeal that well-designed and conducted randomized controlled trials are urgently needed for the treatment of PCNSL patients with the HCT-ASCT method. I recommend the review manuscript for publication in Cancers after the authors address the following issues:
- For all the clinical trials in PCNSL patients, how many studies discuss the statistical analysis, and what results are convincing? The authors need to discuss these since too many factors are included in the review.
- The authors comment that the HCT-ASCT strategy in elderly PCNSL patients is scarce but may be promising. However, high toxicity, an infectious rate even mortality rate may be also associated with the trials since one study was terminated due to a 16.7% mortality rate. Do the authors think these are a potential risk/concern for the few trials in elderly populations or just because of the limited number of randomly enrolled elderly patients?
- Please pay attention to several typos, such as line 106 “(toxicities))”, “2 28-day-cycles”, “3 28-day-cycles”, “6 14-day cycles”, “scare”.
- Please update and double-check the references such as refs. 70, 73, 94.
